# Photosynthetic and Morphological Acclimation to High and Low Light Environments in *Petasites japonicus* subsp. *giganteus*

**Ray Deguchi and Kohei Koyama ***

Laboratory of Plant Ecology, Department of Life Science and Agriculture, Obihiro University of Agriculture and Veterinary Medicine, Inada-cho, Obihiro, Hokkaido 080-8555, Japan; obihiro.plant.ecology.students2@gmail.com
* Correspondence: koyama@obihiro.ac.jp

**Abstract:** Within each species, leaf traits such as light-saturated photosynthetic rate or dark respiration rate acclimate to local light environment. Comparing only static physiological traits, however, may not be sufficient to evaluate the effects of such acclimation in the shade because the light environment changes diurnally. We investigated leaf photosynthetic and morphological acclimation for a perennial herb, butterbur (*Petasites japonicus* (Siebold et Zucc.) Maxim. subsp. *giganteus* (G.Nicholson) Kitam.) (Asteraceae), in both a well-lit clearing and a shaded understory of a temperate forest. Diurnal changes in light intensity incident on the leaves were also measured on a sunny day and an overcast day. Leaves in the clearing were more folded and upright, whereas leaves in the understory were flatter. Leaf mass per area (LMA) was approximately twofold higher in the clearing than in the understory, while light-saturated photosynthetic rate and dark respiration rate per unit mass of leaf were similar between the sites. Consequently, both light-saturated photosynthetic rate and dark respiration rate per unit area of leaf were approximately twofold higher in the clearing than in the understory, consistent with previous studies on different species. Using this experimental dataset, we performed a simulation in which sun and shade leaves were hypothetically exchanged to investigate whether such plasticity increased carbon gain at each local environment. As expected, in the clearing, the locally acclimated sun leaves gained more carbon than the hypothetically transferred shade leaves. By contrast, in the understory, the daily net carbon gain was similar between the simulated sun and shade leaves on the sunny day due to the frequent sunflecks. Lower LMA and lower photosynthetic capacity in the understory reduced leaf construction cost per area rather than maximizing net daily carbon gain. These results indicate that information on static photosynthetic parameters may not be sufficient to evaluate shade acclimation in forest understories.

**Keywords:** phenotypic plasticity; shade tolerance; shade acclimation; light acclimation; light regime; sunfleck; leaf thickness; leaf angle; leaf three-dimensional structure

## 1. Introduction

In forests, individual plants from a single species often experience various light environments, from well-lit clearings or large gaps to shaded understories [1–4]. For plants, as sessile organisms, phenotypic plasticity is essential for survival in such heterogeneous environments [3,5–8]. This phenotypic plasticity and the consequent intraspecific variation also greatly influence community-level plant traits and productivity [9–15], highlighting the importance of the quantification of phenotypic plasticity of plant traits under different light environments.

In shaded understories, maximizing net carbon gain [3,6,16–18] and maximizing stress tolerance [19–22] are two major determinants of plant survival [18,23]. For maximizing net

photosynthetic carbon gain, acclimation of leaf physiological traits [24,25] and biomass allocation patterns [18,24,25] are both important strategies. Within a species, plants grown in shaded places have leaves with a lower light-saturated photosynthetic rate [1–7,15,18–20,26–31] and a lower dark respiration rate [1–7,18], have thinner leaves with a lower leaf mass per unit area associated with their lower biomass investment per unit area [4,15,18–20,27–29,32–34], and have a higher leaf mass ratio (i.e., leaf mass relative to whole-plant mass) [5,18,25,28] than plants grown in well-lit places. Analogous leaf acclimation to different light environments has also been reported for sunlit and shaded leaves within a single canopy or within a single plant [26,29,34–47]. A low dark respiration rate of a shade leaf leads to a lower photosynthetic light compensation point (LCP) [3,5,6,17,18,24,30,48]. It has been frequently suggested that the net daily carbon gain would increase by lowering the LCP in the shade [3,17,24]. Such a simple consideration, however, has limitations because it only evaluates static photosynthetic parameters. In the forest understory, light intensity changes diurnally due to the diurnal elevation of the sun and fluctuates dynamically due to sunflecks [2,49–60]. A comparison of only static photosynthetic parameters, such as light-saturated photosynthetic rates and dark respiration rates, may therefore poorly reflect actual daily photosynthesis in field environments [42,51,56,61,62]. Given this, it has been questioned whether simple sun vs. shade acclimation can be understood based on the steady-state photosynthetic rate [23,42]. Additionally, the results of laboratory experiments under controlled low light environments [6,20] or those of field shading experiments using shade cloths [19,30,33] may not provide an accurate estimate of carbon gain in the understory, because they do not take into consideration sunflecks. To understand the effect of shade acclimation on daily net carbon gain, therefore, the effects of sunflecks also should be considered.

Here, we investigated the shade acclimation of *Petasites japonicus* subsp. *giganteus* that naturally grew in either well-lit or shaded places in a temperate forest. In a previous study on the same species [32], the phenotypic plasticity of some leaf traits under different light environments was reported. However, because the authors did not measure photosynthetic parameters and local light intensity, they did not clarify whether such plasticity contributed to maximizing carbon gain under each light environment. The objectives of this study, therefore, were (1) to quantify the photosynthetic and morphological acclimation to different light environments for this species, and (2) to test whether leaf physiological acclimation contributed to maximizing leaf-level carbon gain under diurnally changing light environment due to sunflecks.

## 2. Materials and Methods

### 2.1. Study Species

Butterbur (*Petasites japonicus* (Siebold et Zucc.) Maxim. subsp. *giganteus* (G.Nicholson) Kitam.) (Asteraceae) is a perennial herb distributed in Northeast Asia [63]. This species is found naturally in environments of varying amounts of light, such as roadsides, well-lit forest gaps, and in shaded forest understories. This species also is grown as a vegetable in eastern Asia, including Japan, Korea [64], and Taiwan [65]. Large radical leaves (often reaching 1–2 m in height) elongate from an underground shoot in this species (Figure 1a–c). Therefore, investigating the leaves is equivalent to investigating the entire above-ground part (ramet) for this clonal species. These leaves are usually horizontally arranged on the ground so as to prevent overtopping others, but small immature leaves that are not fully expanded often exist below fully expanded leaves.

### 2.2. Study Site and Sampling

We performed the study at two sites in the same forest (clearing [C] and understory [U]), which were approximately 100 m apart, in the Forest of Obihiro (Obihironomori) (42°53′ N, 143°09′ E, altitude: 86 m a.s.l.). This secondary forest comprises a mixture of planted and regenerated trees and is located in Obihiro City in eastern Hokkaido in a cool-temperate region in Japan. The mean annual temperature and precipitation at the Japan Meteorological Agency Obihiro Weather Station (6 km from the study site)

between 1998 and 2017 were 7.2 °C and 937 mm, respectively [66]. In the clearing site (approximately 30 × 30 m), few trees were taller than the investigated leaves (Figure 2a). The understory site (Figure 2b) (approximately 15 × 10 m) was located under a young birch forest (*Betula platyphylla* Sukaczev var. *japonica* (Miq.) H.Hara; DBH: 17.5–25.1 cm), in which some walnut (*Juglans mandshurica* Maxim. var. *sachalinensis* (Komatsu) Kitam.) grew as subcanopy trees. Within each plot, the investigated leaves were selected along a transect. Although we found multiple separate patches of leaves at each site, the number of genets was unknown. Therefore, the investigated leaves were selected as evenly as possible along the entire length of each transect.

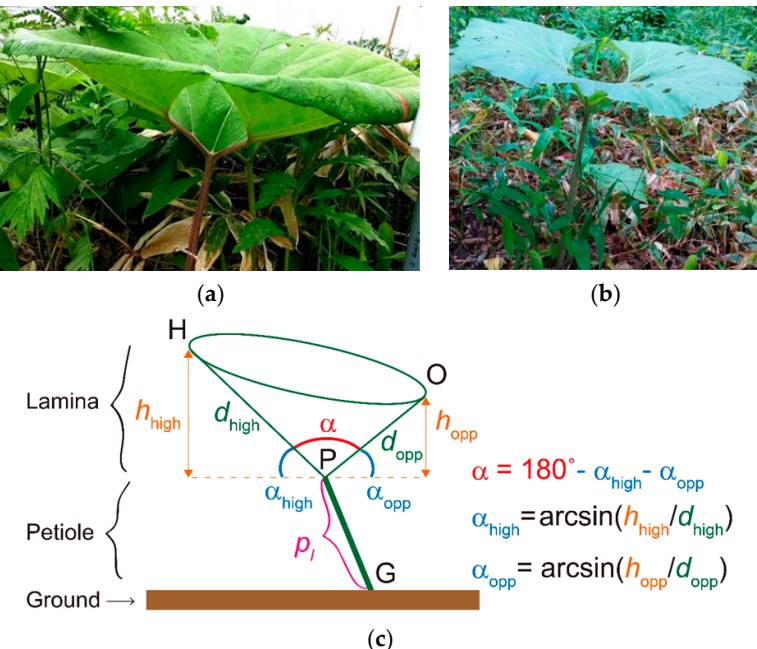

**Figure 1.** The measured morphological parameters of the leaves of the butterburs (*Petasites japonicus* (Siebold et Zucc.) Maxim. subsp. *giganteus* (G.Nicholson) Kitam.) investigated in this study. Leaves in (**a**) the clearing and (**b**) the understory, in addition to (**c**) the measured leaf parameters, are shown. H: The highest point on the leaf lamina. O: The point located at the opposite side of H on the lamina edge. P: The point of attachment of the lamina to the petiole. G: The point of attachment of the petiole to the ground. $d_{high}$: The distance between H and P. $h_{high}$: The vertical distance between H and P. $d_{opp}$: the distance between O and P. $h_{opp}$: the vertical distance between O and P. $p_l$: Above-ground petiole length (the distance between P and G). $\alpha$: Lamina openness angle. Photographs were taken in June 2020 by Kohei Koyama.

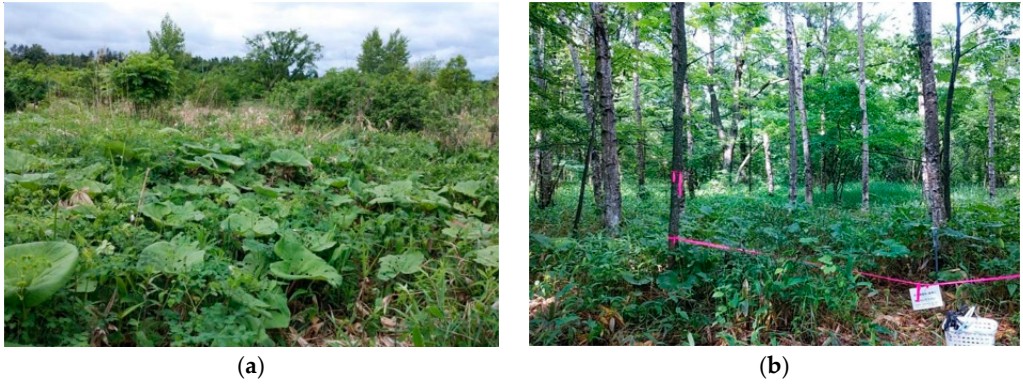

**Figure 2.** The study sites, (**a**) the clearing and (**b**) the understory, in the Forest of Obihiro. Photographs were taken on (**a**) June 28 and (**b**) 3 July 2020 by Kohei Koyama.

In June 2020, we marked 62 leaves (32 from plants in the clearing and 30 from plants in the understory). Leaf three-dimensional structure was measured using measuring tapes on 24–25 June 2020, and the lamina openness angle [67,68] was calculated (Figure 1c). Photosynthetic light response curves were measured for a total of 12 leaves (6 at each site) on 21, 22 and 24 June 2020 with a portable photosynthesis system (LI-6400; LI-COR, Lincoln, NE, USA) equipped with an LED light source (LI-6400-02B) (Figure 3a). Due to the amount of rainfall prior to the measurement days (June 18 (3 mm), June 19 (7.5 mm), June 20 (4 mm), and June 23 (1 mm), data from [66]), the soil in the fields was wet during the measurements. Measurements were taken in the morning (7:30–12:00) each day under cloudy and humid conditions, and the environment inside the chamber showed favorable conditions for photosynthesis: leaf temperature (measured by a thermocouple inside the chamber) ranged from 17.93 to 24.48 °C, and the vapor pressure deficit (VPD) based on leaf temperature was always less than 0.9 kPa. In the understory, we first induced the leaves by keeping incident photosynthetic photon flux density (PPFD) on the leaf surface at 1000–1500 $\mu$mol m$^{-2}$ s$^{-1}$ until equilibration. This process was omitted for most of the leaves in the clearing, which showed a very quick response under PPFD 2000 $\mu$mol m$^{-2}$ s$^{-1}$. Then, we progressively lowered the incident PPFD on the leaf surface (2000, 1500, 1000, 750, 500, 250, 125, 63, 32, and 0 $\mu$mol m$^{-2}$ s$^{-1}$). On each occasion of changing light intensity, we kept the PPFD constant until the equilibration of the leaves. The $CO_2$ concentration of the air entering the leaf chamber was controlled at 400 ppm. All the data recorded by the LI-6400 (e.g., photosynthesis, stomatal conductance, transpiration, humidity, temperature at each PPFD, etc.) are available in the Supplementary Materials.

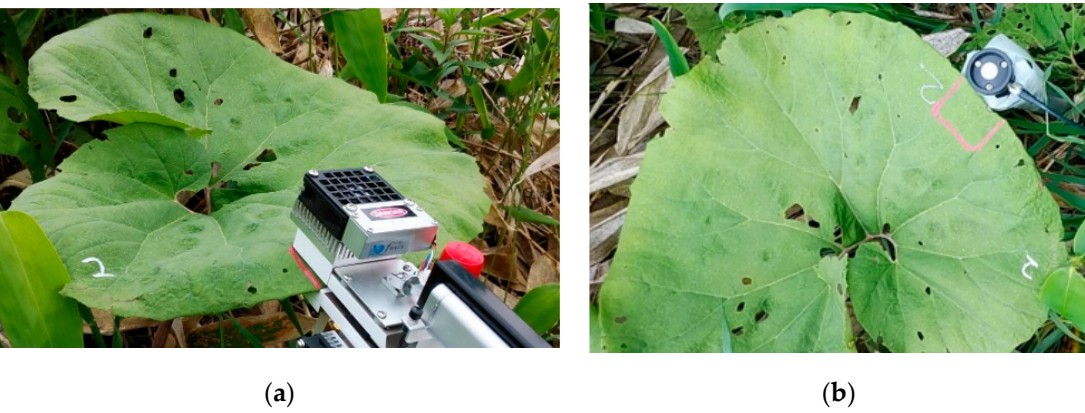

**(a)**　　　　　　　　　　　　　　　　　　　　　　**(b)**

**Figure 3.** Measurements of (**a**) photosynthesis and (**b**) the diurnal course of incident light in the clearing site. The two panels show the same leaf. The position on the leaf lamina, at which photosynthetic traits and incident light were measured, was marked with a light-resistant ink pen (red box). In the cases when that part of the lamina was inclined, the light sensor was inclined such that the lamina and the top of the sensor were parallel to one another. Leaf mass per area (LMA) was subsequently measured by sampling the lamina part within the same red box. Photographs were taken on (**a**) June 21 and (**b**) 1 July 2020 by Kohei Koyama.

Net photosynthetic rate per unit area of each leaf ($P_{net}$ $\mu$mol m$^{-2}$ s$^{-1}$) was assumed to be expressed by the non-rectangular hyperbola (NRH) [69]:

$$P_{net} = \frac{\Phi I + P_{g\_max\_area} - \sqrt{(\Phi I + P_{g\_max\_area})^2 - 4\theta \Phi I P_{g\_max\_area}}}{2\theta} - R_{area}, \quad (1)$$

where $I$ ($\mu$mol m$^{-2}$ s$^{-1}$) indicates the incident PPFD for each leaf at each moment, and $P_{g\_max\_area}$ ($\mu$mol m$^{-2}$ s$^{-1}$) indicates the maximum gross photosynthetic rate when $I$ approaches infinity. $\Phi$ (mol $CO_2$ mol$^{-1}$ quanta) and $\theta$ (dimensionless) indicate the initial slope and the convexity, respectively. $R_{area}$ ($\mu$mol m$^{-2}$ s$^{-1}$) indicates the dark respiration rate. These parameters were fitted for each leaf

with the R function *nls*. Light compensation point (LCP) was calculated by solving the quadratic form of NRH [68] for *I* on the condition that $P_{net} = 0$ [70] using the software Maxima (Maxima project, USA) [71]:

$$\theta(Pnet + Rarea)^2 - (\phi I + Pg\_max\_area)(Pnet + Rarea) + \phi IPg\_max\_area = 0$$
$$Pnet = 0 \Rightarrow I \equiv LCP = \frac{Rarea(Rarea\theta - Pg\_max\_area)}{(Rarea - Pg\_max\_area)\Phi} \tag{2}$$

### 2.3. Measurement of PPFD

#### 2.3.1. Diurnal Course of Incident PPFD on the Leaves

We measured a time-series of incident PPFD on the selected leaves on two days: an overcast day (June 24; clearing, *n* = 4; understory, *n* = 4) and a sunny day (July 3; clearing, *n* = 3; understory: *n* = 4) in 2020. The sample size difference between these two days was due to a measurement failure caused by an operational error. On the days between June 24 and July 3, PPFD data were not obtained due to intermittent disruptions by rain. Those leaves were selected from the samples for which photosynthetic light response curves were measured. PPFDs were measured for the same parts of the leaves as for the photosynthetic parameters (the red box, Figure 3). For each target leaf, we set one quantum sensor (MIJ-14PAR Type2/K2; Environmental Measurement Japan, Fukuoka, Japan) on the pole. If the leaf part was inclined, the sensor was inclined to measure the incident PPFD on the leaf surface (Figure 3b). Each sensor was connected to a voltage logger (LR5041; HIOKI, Ueda, Japan). Voltage was recorded every 10 min for 24 h. These voltage values were transformed into PPFD using sensor-specific coefficients.

#### 2.3.2. Instantaneous Measurement of rPPFD

In addition to measuring the diurnal course of PPFD for the selected leaves, we measured instantaneous PPFDs for all 62 leaves to estimate the light environment at the two sites. The measurements were conducted around midday on an overcast day (20 June 2020). We used two quantum sensors (MIJ-14PAR) and simultaneously measured the PPFD incident on each leaf and on top of an approximately 2 m tripod placed at the open place. One sensor at the tripod was fixed horizontally. The other was placed at a tip of a hand-held measuring bar [4,72] and was inclined for each leaf to measure the PPFD at the bottom surface of the cone- or funnel-shaped leaf (Figure 4). Each sensor was connected to a voltage logger (LR5041), and the voltages of the two sensors were recorded simultaneously. We calculated the relative PPFD (rPPFD) as the ratio of the PPFD on each leaf to the PPFD at the top of the tripod.

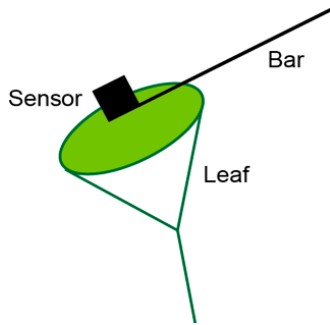

**Figure 4.** A schematic diagram of instantaneous relative photosynthetic photon flux density (rPPFD) measurement with a hand-held measuring bar (illustration by Kohei Koyama).

### 2.4. Leaf Thickness and LMA

On 4 and 7 July 2020, we sampled pieces (ca. 40 cm$^2$) from the lamina edges of 41 leaves (clearing, *n* = 20; understory, *n* = 21), measured their thickness with a digital caliper (CD-15PSX (resolution:

0.01 mm); Mitutoyo Corp, Kawasaki, Japan), and scanned them immediately after sampling with an A4 flatbed scanner (CanoScan LiDE 220; Canon, Tokyo, Japan). These samples included all 12 leaves for which photosynthetic rates were measured, and the lamina samples were taken at the same positions on the leaves as the photosynthetic measurements (Figure 3). The lamina parts were selected to avoid the thickest leaf veins. The laminae were then oven-dried at 70 °C for at least one week, and their dry mass was measured with a precision balance. The projected area of each piece of lamina was measured with the Image J software (NIH, Bethesda, MD, USA) [73]. We calculated leaf mass per area (LMA, g m$^{-2}$) as the ratio between the dry mass and the area of one side of each sampled piece of lamina [74]. We calculated mass-based values by dividing the area-based values by the LMA of that leaf [2].

## 2.5. Calculation of Daily Photosynthesis

We calculated the instantaneous net photosynthetic rate for each target leaf every 10 min for 24 h using the estimated light response curves (Equation (1)) and the diurnal course of incident PPFD. At night, the dark respiration rate ($R_{area}$) was used as the nighttime respiration rate. By integrating these, we calculated the daily net photosynthesis for each target leaf.

## 2.6. Simulation 1: Exchanged Leaves

We performed a simulation in which we hypothetically exchanged the photosynthetic light response curves between the two sites. We expected that if the difference in the photosynthetic traits between the sites was the acclimation to a local light environment, then the exchange of leaves would reduce the daily net photosynthesis for both sites. For each understory leaf, we replaced the photosynthetic light response parameters ($P_{g\_max\_area}$, $R_{area}$, $\Phi$, and $\theta$ in Equation (1)) with the mean photosynthetic parameters obtained in the clearing and vice versa. Next, we calculated the daily net photosynthesis for each hypothetical leaf by using the actual incident PPFD on each leaf. We also calculated the critical PPFD for each understory leaf. This value was defined as the lowest PPFD, such that if instantaneous PPFD exceeded that value, the net photosynthetic rate at that moment would be higher for the hypothetically set clearing leaf than for the actual understory leaf in question. This value was used to investigate how often PPFD exceeded this value due to sunflecks.

## 2.7. Simulation 2: Understory without Sunflecks

To evaluate the significance of sunflecks in the understory, we performed another simulation in which sunflecks were hypothetically removed from the original dataset of the diurnal course of PPFD on the sunny day (July 3). If PPFD at a given moment exceeded 200 µmol m$^{-2}$ s$^{-1}$, that PPFD value was replaced by a fixed value of 200 µmol m$^{-2}$ s$^{-1}$. This value was approximately equal to the maximum PPFD (199.72) observed in the understory on the overcast day (June 24). We also observed that the diel cycle of PPFD in the understory did not exceed 200 µmol m$^{-2}$ s$^{-1}$ on the sunny day except during sunflecks (see Results), so that the background diel cycle of PPFD was retained by this simulation. We then calculated the daily net photosynthesis in the understory without sunflecks, either with (1) actual understory leaves, or (2) the hypothetically set clearing leaf (described above).

## 2.8. Statistical Analysis

All statistical analyses were performed with the statistical software R v4.0.3 (Vienna, Austria) [75] and packages ("cowplot" [76], "ggbeeswarm" [77], "ggplot2" [78], and "lme4" [79]). The results were compared between sites by fitting generalized linear models (GLM) using the function *glm* (family = Gamma (link = "log")), except for the simulation results. The differences in the simulation results obtained under different scenarios were tested using a generalized linear mixed model (GLMM), treating individual leaves (i.e., diurnal courses of PPFD on different leaves) as random intercepts, and using the function *glmer* (family = Gamma (link = "log")) [79], except in one case (simulated clearing leaves in the understory on the overcast day), in which simulated net daily photosynthesis

values contained a negative value. For that case, a linear mixed model (LMM) was fit with the function *lmer* [79]. In all cases, the significance of the fixed effect was tested using the likelihood ratio test with the function *anova* (test = "Chisq").

## 3. Results

### 3.1. Leaf Shape

The differences in the light environment were quantified by large differences in rPPFD and daily light integral between the two sites (Table 1). Lamina openness angle was significantly larger in the understory (U) than in the clearing (C) ($p < 0.01$, Table 1; Figure 5a,b), indicating that the three-dimensional arrangement of leaf laminae was flatter in the understory (see photographs in Figure 1a,b). Compared with the difference in shape, the difference in lamina size was relatively small: $d_{high}$ was slightly larger in the understory than in the clearing ($p = 0.025$), and neither $d_{opp}$ ($p = 0.37$) nor the petiole length ($p_l$) ($p = 0.28$) significantly differed between the sites (Table 1; Figure 5c–e).

**Table 1.** Leaf traits in the clearing (C) and in the understory (U).

| Symbol | Definition | Unit | Sample Size ($n$) C | Sample Size ($n$) U | Mean (C) | Mean (U) | Ratio C/U | C Vs. U *P*-Value |
|---|---|---|---|---|---|---|---|---|
| rPPFD | - | - | 32 | 30 | 87.7% | 10.8% | 8.10 | <0.01 |
| $\alpha$ | Lamina openness angle | Degree (°) | 32 | 30 | 116.3 | 136.5 | 0.85 | <0.01 |
| $d_{high}$ | Lamina length (high) | cm | 32 | 30 | 31.3 | 35.2 | 0.89 | 0.025 |
| $d_{opp}$ | Lamina length (opposite) | cm | 32 | 30 | 29.7 | 31.1 | 0.96 | 0.38 ns |
| $p_l$ | Petiole length | cm | 32 | 30 | 55.1 | 57.7 | 0.95 | 0.28 ns |
| LMA | Leaf mass per area | g m$^{-2}$ | 20 | 21 | 44.3 | 23.7 | 1.87 | <0.01 |
| - | Leaf thickness | mm | 20 | 21 | 0.428 | 0.348 | 1.23 | <0.01 |
| $P_{net\_2000}$ | Net photosynthetic rate at PPFD = 2000 mol m$^{-2}$ s$^{-1}$ | µmol m$^{-2}$ s$^{-1}$ | 6 | 6 | 23.2 | 11.3 | 2.04 | <0.01 |
| $P_{g\_max\_area}$ | Maximum gross photosynthetic rate per unit leaf area | µmol m$^{-2}$ s$^{-1}$ | 6 | 6 | 27.2 | 12.6 | 2.16 | <0.01 |
| $R_{area}$ | Dark respiration rate per unit leaf area | µmol m$^{-2}$ s$^{-1}$ | 6 | 6 | 1.85 | 0.87 | 2.14 | <0.01 |
| $\Phi$ | Initial slope | mol CO$_2$ mol$^{-1}$ quanta | 6 | 6 | 0.075 | 0.076 | 0.98 | 0.48 ns |
| $\theta$ | Convexity | - | 6 | 6 | 0.562 | 0.600 | 0.94 | 0.49 ns |
| LCP | Light compensation point | µmol quanta m$^{-2}$ s$^{-1}$ | 6 | 6 | 25.7 | 11.8 | 2.18 | <0.01 |
| $P_{g\_max\_mass}$ | Maximum gross photosynthetic rate per unit leaf mass | nmol g$^{-1}$ s$^{-1}$ | 6 | 6 | 545 | 523 | 1.04 | 0.72 ns |
| $R_{mass}$ | Dark respiration rate per unit leaf mass | nmol g$^{-1}$ s$^{-1}$ | 6 | 6 | 37.1 | 36.7 | 1.01 | 0.97 ns |
| $I_{day}$ | Daily light integral—sunny day | mol quanta m$^{-2}$ day$^{-1}$ | 3 | 4 | 45.7 | 5.96 | 7.67 | <0.01 |
| | —overcast day | | 4 | 4 | 26.5 | 2.74 | 9.69 | <0.01 |
| $P_{n\_day}$ | Net daily photosynthesis per area—sunny day | mol m$^{-2}$ day$^{-1}$ | 3 | 4 | 0.745 | 0.145 | 5.14 | <0.01 |
| | —overcast day | | 4 | 4 | 0.682 | 0.075 | 9.05 | <0.01 |
| $R_{day}$ | Daily respiration per area | mol m$^{-2}$ day$^{-1}$ | 6 | 6 | 0.160 | 0.075 | - [1] | - [1] |
| $P_{g\_day}$ | Gross daily photosynthesis per area—sunny day | mol m$^{-2}$ day$^{-1}$ | 3 | 4 | 0.906 | 0.237 | 3.83 | <0.01 |
| | —overcast day | | 4 | 4 | 0.841 | 0.167 | 5.03 | <0.01 |

[1] Same values as $R_{area}$.

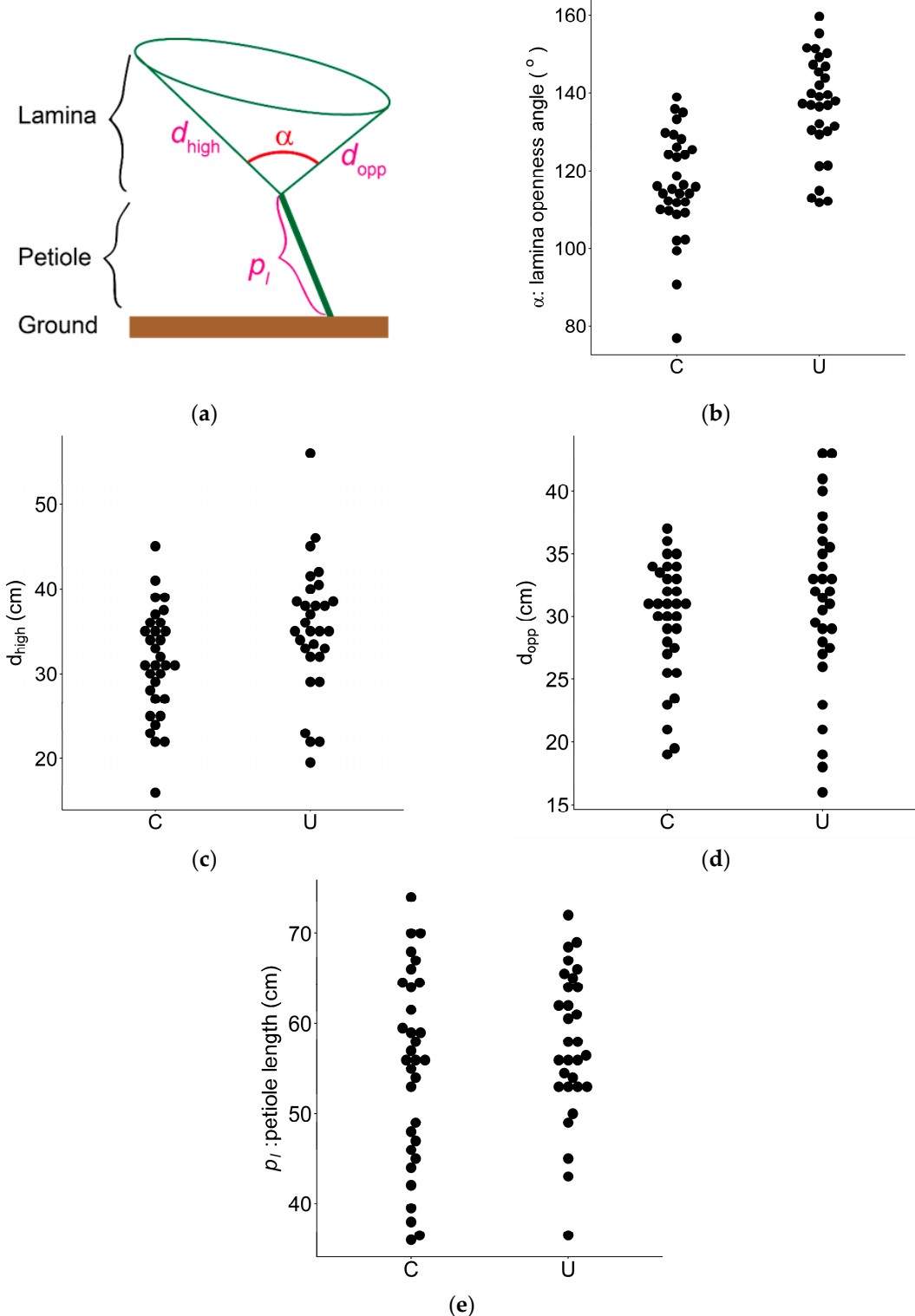

**Figure 5.** (**a**–**e**) Leaf shape parameters in the understory (U) and in the clearing (C). Each closed circle indicates one leaf (i.e., bee swarm plot).

*3.2. Area-Based Photosynthetic Traits*

The area-based net photosynthetic rate at PPFD = 2000 μmol m$^{-2}$ s$^{-1}$ ($P_{net\_2000}$), the maximum gross photosynthetic rate ($P_{g\_max\_area}$), dark respiration rate ($R_{area}$), and light compensation point (LCP) were all significantly higher in the clearing than in the understory ($p < 0.01$) (Table 1; Figure 6a–d).

Neither the initial slope ($\Phi$) nor the convexity ($\theta$) significantly differed between the sites ($p$ = 0.48–0.49) (Table 1; Figure 6e,f).

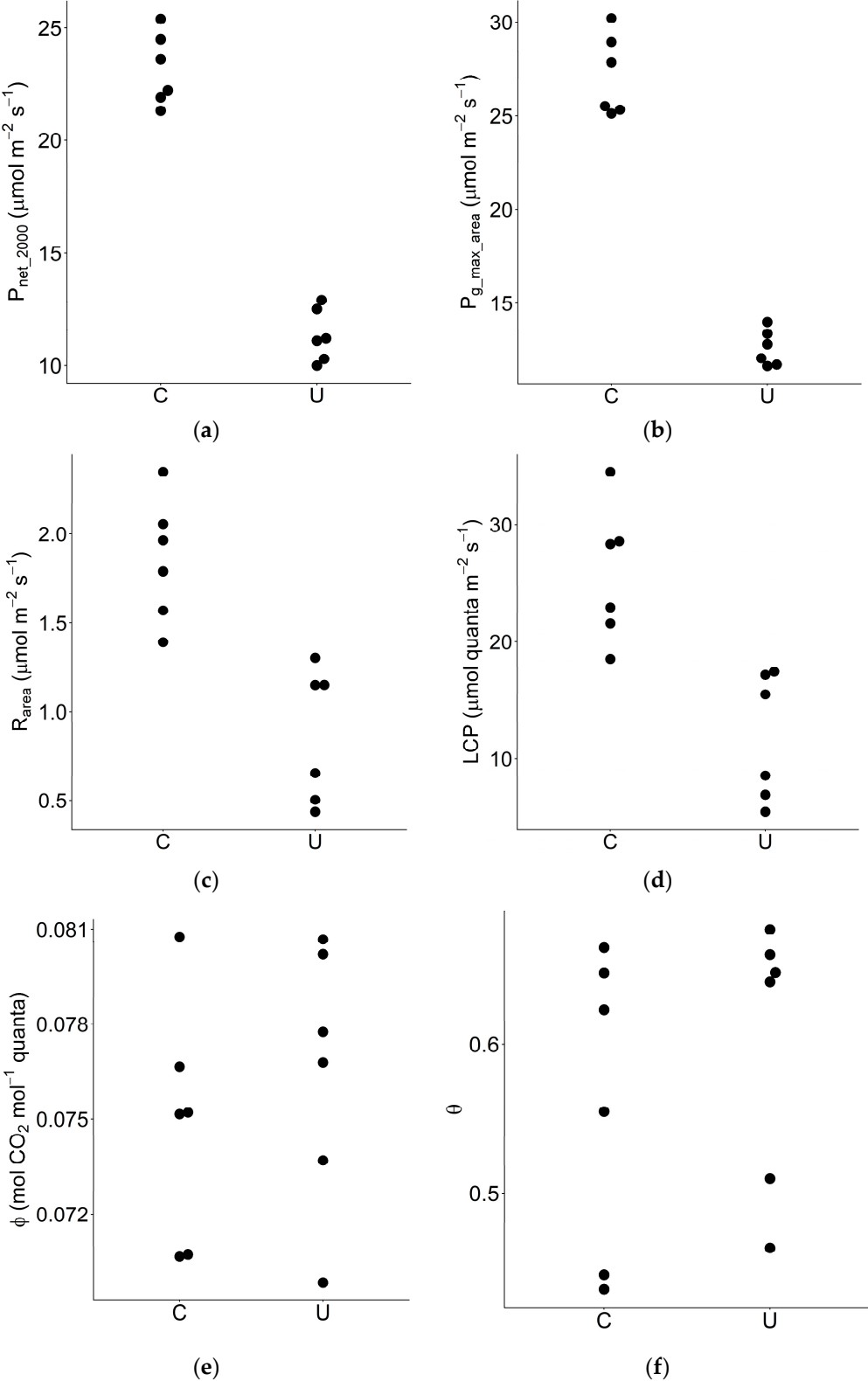

**Figure 6.** (**a–f**) Area-based photosynthetic parameters of the sun leaves in the clearing (C) and the shade leaves in the understory (U). Each closed circle indicates one leaf. LCP: light compensation point.

### 3.3. LMA and Leaf Thickness

The mean LMA of the leaves in the clearing leaf was 1.87 times larger than that of the leaves in the understory ($p < 0.01$) (Table 1; Figure 7a). Additionally, the mean leaf thickness was 1.23 times larger than that of the leaves in the understory ($p < 0.01$) (Table 1; Figure 7b).

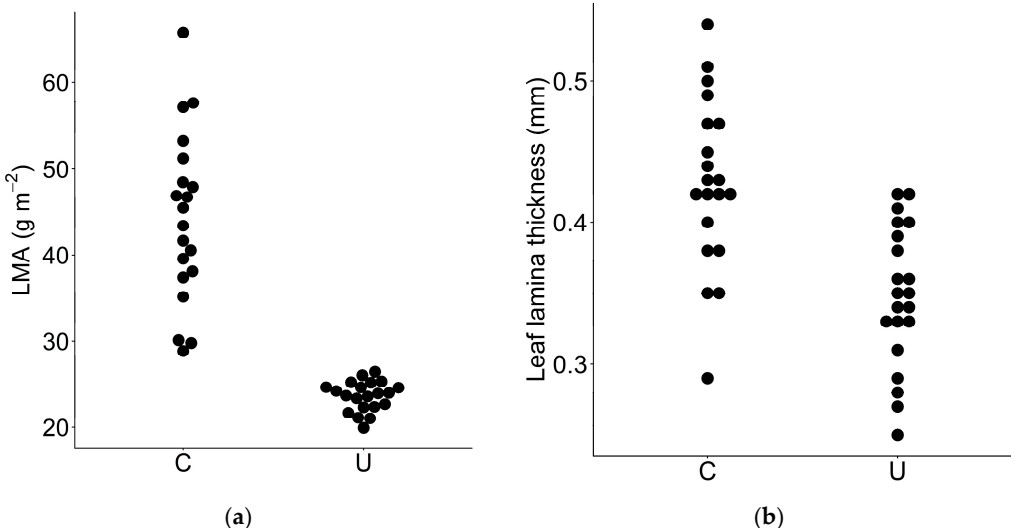

**Figure 7.** (**a**) Leaf mass per unit area (LMA) and (**b**) lamina thickness of the sun leaves in the clearing (C) and the shade leaves in the understory (U). Each closed circle indicates one leaf. The thicknesses were measured with a resolution of 0.01 mm.

### 3.4. Mass-Based Photosynthetic Traits

In contrast to the high plasticity in the area-based rates, no significant difference was found for mass-based photosynthetic and respiration rates between the sites ($p = 0.69$–$0.96$) (Table 1; Figure 8a,b).

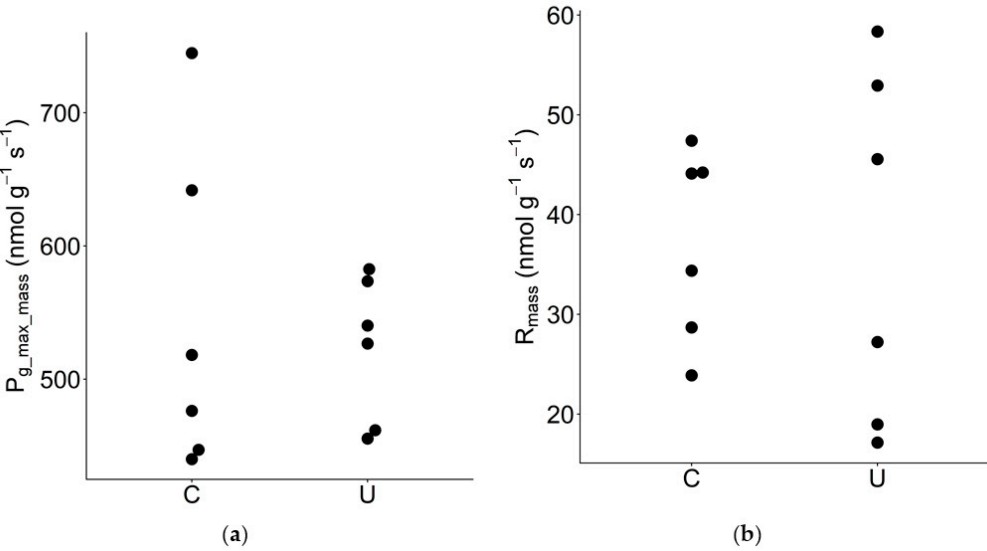

**Figure 8.** Mass-based (**a**) photosynthetic rates and (**b**) dark respiration rates of the sun leaves in the clearing (C) and the shade leaves in the understory (U). Each closed circle indicates one leaf.

### 3.5. Diurnal Courses of PPFD

Figure 9 shows the diurnal courses of PPFD incident on the leaves. The estimated critical values were 59–161 $\mu$mol m$^{-2}$ s$^{-1}$ (median: 111.5; these values were calculated for all the six understory leaves

for which photosynthetic parameters were measured, and the diurnal course of PPFD was measured for four of them, as shown in Figure 9b,d). On the sunny day in the understory, instantaneous PPFD often exceeded the critical values due to sunflecks (Figure 9b).

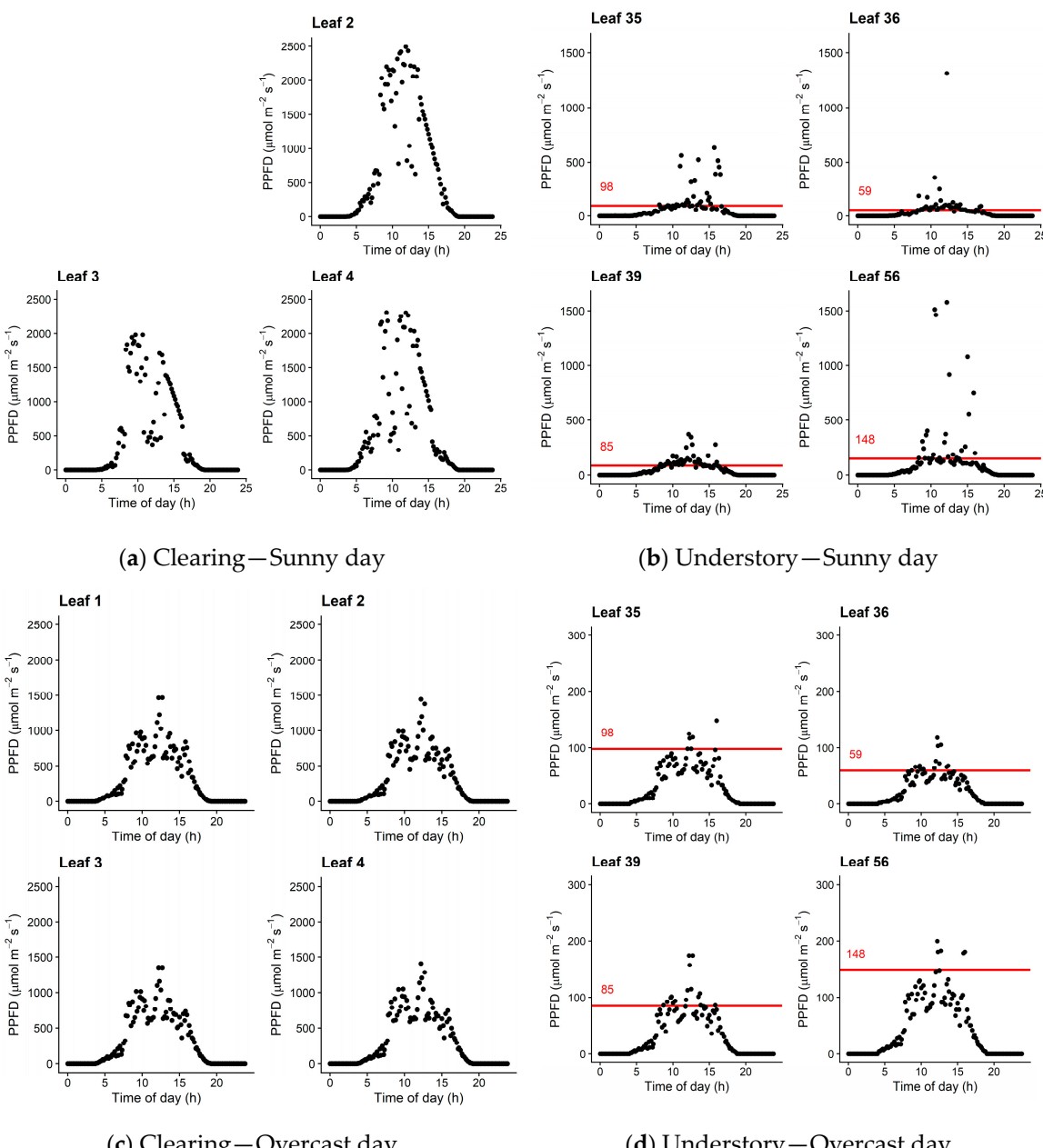

**Figure 9.** Diurnal course of light intensity (PPFD) incident on each leaf at each site (the clearing and understory) on a sunny day (3 July 2020) and on an overcast day (24 June 2020). (**a**) Clearing on the sunny day, (**b**) understory on the sunny day, (**c**) clearing on the overcast day, and (**d**) understory on the overcast day. Data for Leaf 1 on the panel (**a**) were not obtained owing to a measurement failure. The red horizontal lines on the understory panels indicate the critical values of PPFD; if PPFD at one moment exceeded that value, the instantaneous net photosynthetic rate would be higher for a typical clearing leaf than for the understory leaf in question.

*3.6. Daily Carbon Gain under Actual and Simulated Conditions*

The obtained results differed between the two sites and between weather conditions (Table 2). In the clearing, the simulated change from the actual clearing leaves into understory leaves greatly

(>40%) reduced the daily net photosynthesis on both the sunny day (from A to B in Figure 10a; mean: (A) 0.745 and (B) 0.430 mol m$^{-2}$ day$^{-1}$; A vs. B: $p < 0.01$) and on the overcast day (from A to B in Figure 10b; (A) 0.682 and (B) 0.415 mol m$^{-2}$ day$^{-1}$; A vs. B: $p < 0.01$). This was because on both days, the reduction in gross photosynthesis (from C into D in Figure 10a,b) was much larger in magnitude than that of respiratory loss (from E into F in Figure 10a,b).

**Table 2.** Daily carbon exchange per unit area of leaf. The mean value for each item is shown.

| Daily Carbon Exchange | Simulation | Clearing (C) | | Understory (U) | |
|---|---|---|---|---|---|
| | | Sunny | Overcast | Sunny | Overcast |
| Daily net photosynthesis (mol m$^{-2}$ day$^{-1}$) | Sample size | $n = 3$ | $n = 4$ | $n = 4$ | $n = 4$ |
| | Actual leaves | 0.745 | 0.682 | 0.145 | 0.075 |
| | Exchanged leaves (between C and U) | 0.430 | 0.415 | 0.138 | 0.024 |
| | Without sunflecks (actual leaves) | - | | 0.134 | - |
| | Without sunflecks (exchanged leaves) | - | | 0.106 | - |
| Daily gross photosynthesis (mol m$^{-2}$ day$^{-1}$) | Sample size | $n = 3$ | $n = 4$ | $n = 4$ | $n = 4$ |
| | Actual leaves | 0.906 | 0.803 | 0.237 | 0.167 |
| | Exchanged leaves | 0.505 | 0.471 | 0.298 | 0.184 |
| Daily respiration (mol m$^{-2}$ day$^{-1}$) | Sample size [1] | $n = 6$ | | $n = 6$ | |
| | Actual leaves | 0.160 | | 0.075 | |
| | Exchanged leaves | 0.075 | | 0.160 | |

[1] Daily leaf respiration rates were calculated for all the 12 leaves (6 in each site) for which dark respiration rates were measured, and the mean value at each site is shown in Table 2 and used in the leaf-exchange simulation. The same daily respiration rates were used for the two days. Among these 12 leaves, daily net- and gross-photosynthetic rates were measured or simulated for seven or eight selected leaves.

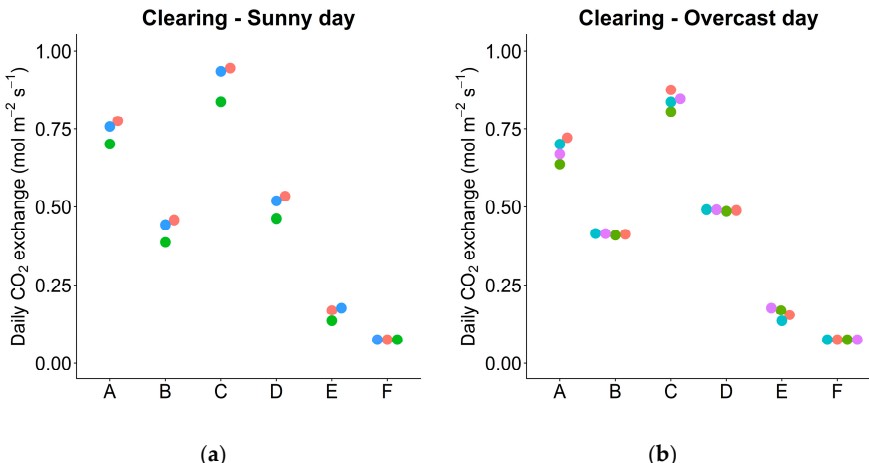

**Figure 10.** Daily carbon exchange per unit area of leaf in the clearing on (**a**) a sunny day (3 July 2020) and (**b**) an overcast day (24 June 2020). Results of the same leaf simulated in different scenarios appear in the same color. A: Estimated actual net photosynthesis. B: Simulated net photosynthesis when the photosynthetic parameters were hypothetically changed to those of the understory. C: Estimated actual gross photosynthesis. D: Simulated gross photosynthesis when the photosynthetic parameters were hypothetically changed to those of the understory. E: Estimated actual daily respiration. F: Simulated daily respiration when the leaves were changed to the understory leaves. In simulation F, all three or four leaves were assumed to have the same respiration rate as the mean value of the understory leaves.

By contrast, in the understory, the simulated change from actual understory leaves to the clearing leaves did not greatly reduce the daily net carbon gain on the sunny day (Table 2; from A to B, Figure 11a; (A) 0.145 and (B) 0.138 mol m$^{-2}$ day$^{-1}$; A vs. B: $p = 0.082$). This is because on the sunny day, during which frequent sunflecks were observed (Figure 9b), the increment of gross daily photosynthesis (from C into D in Figure 11a) had approximately the same magnitude as the increment of respiratory loss (from E to F in Figure 11a). Those two effects offset each other. On the overcast day, during which few sunflecks were observed (Figure 9d), the simulated change from the actual understory leaves to the clearing leaves reduced the daily net photosynthesis (from A to B, Figure 11b; mean: (A) 0.075 and (B) 0.024 mol m$^{-2}$ day$^{-1}$; A vs. B: $p < 0.01$).

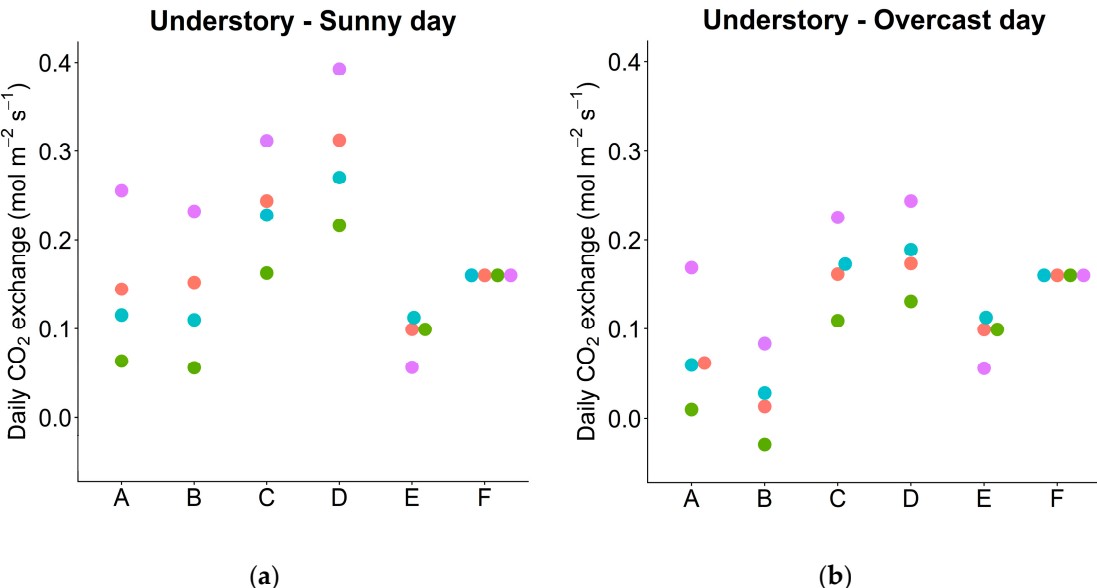

**Figure 11.** Daily carbon gain per unit area of leaf in the understory on (**a**) a sunny day (3 July 2020) and (**b**) an overcast day (24 June 2020). Results of the same leaf simulated in different scenarios appear in the same color. A: Estimated actual net photosynthesis. B: Simulated net photosynthesis when the leaves were hypothetically changed to clearing leaves. C: Estimated actual gross photosynthesis. D: Simulated gross photosynthesis when the leaves were changed to the clearing leaves. E: Estimated actual daily respiration. F: Simulated daily respiration when the leaves were changed to the clearing leaves. In simulation F, all four leaves were assumed to have the same respiration rate as the mean value of the clearing leaves.

### 3.7. Simulation: Understory without Sunflecks

We further examined whether the obtained differences could be explained by the effect of sunflecks. We simulated the net daily carbon gain in the understory on the same sunny day (July 3) under a hypothetical situation in which all sunflecks' PPFD values (>200 μmol m$^{-2}$ s$^{-1}$) were replaced by a fixed value 200 μmol m$^{-2}$ s$^{-1}$. Without sunflecks, the actual understory leaves indeed performed slightly better than the simulated clearing leaves in the understory (Figure 12; mean: (A) 0.134 and (B) 0.106 mol m$^{-2}$ day$^{-1}$; A vs. B: $p < 0.01$).

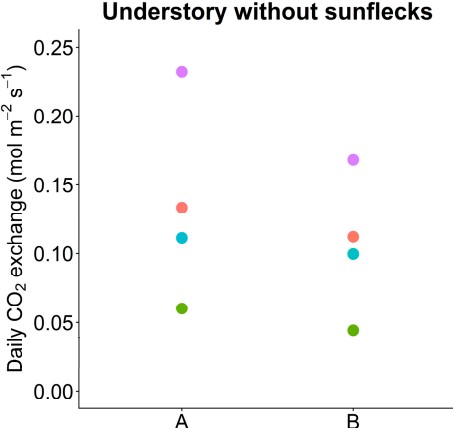

**Figure 12.** Simulated daily carbon gain without sunflecks on the sunny day (3 July 2020), in which sunflecks (>200 µmol m$^{-2}$ s$^{-1}$) were replaced by the fixed value 200 µmol m$^{-2}$ s$^{-1}$. A: Net daily photosynthesis calculated with actual leaves in the understory. B: Net daily photosynthesis when the leaves were hypothetically changed to the clearing leaves. The same leaf simulated under the two different scenarios appears as the same color.

## 4. Discussion

### 4.1. Carbon Gain or Saving via Acclimation

It is frequently discussed that photosynthetic acclimation increases daily net photosynthesis [3,6,17,24,48]. Supporting this theory, net daily carbon photosynthesis per unit leaf area in the clearing was higher for the actual sun leaves than for the simulated shade leaves (Figure 10). This was because during the daytime hours, PPFD on the leaves was always higher than the critical values, irrespective of the diurnal changes or weather (Figure 9a,c). By contrast, in the understory, our results did not always support the same theory; when sunflecks are present, photosynthetic shade acclimation may not always increase net daily photosynthesis. The understory leaves performed better than the clearing leaves in the understory on the overcast day, but not on the sunny day (Figure 11). Lower dark respiration rates in shade-acclimated leaves indeed resulted in lower LCPs (Table 1), but this does not necessarily imply that the shade leaves had higher net carbon gain than the sun leaves in the understory. In the understory, the simulated sun leaves frequently outperformed the shade leaves during the sunflecks (Figure 9b). When these sunflecks were hypothetically removed (Figure 12), or on the overcast day during which few sunflecks were observed (Figure 9d), such shade-acclimated understory leaves indeed had higher net carbon gain than the clearing leaves in the understory light environment. These results suggest that the observed difference was caused by sunflecks. Our results therefore support the notion [23] that under a diurnally fluctuating light environment, information on the static photosynthetic parameters and daily averaged light environment may not be sufficient to evaluate shade acclimation in forest understories. The implication may therefore be that laboratory experiments under controlled light [6,20] or field experiments using shade cloths [19,30,33], in which sunflecks were not taken into consideration, may not provide an accurate estimate of carbon gain in the understory. In this regard, it is possible to obtain better estimates through experiments using plants grown in natural conditions, as observed in [2,60,62] and the present study; those using natural canopy shading [18]; or those using advanced techniques that allow the rapid fluctuation of artificial light intensity in the case of a laboratory experiment [51,80].

For the present case, the observed reduced LMA in the understory can instead be interpreted as an effective cost-saving strategy [24,25,81] rather than as maximizing net daily photosynthesis in low-light environments. LMA was approximately twofold larger in the clearing than in the understory, whereas mass-based photosynthetic capacity ($P_{g\_max\_mass}$) and respiration rate ($R_{mass}$) were similar between the sites. Consequently, both light-saturated photosynthetic capacity and respiration rate per unit area of

the leaves ($P_{g\_max\_area}$ and $R_{area}$) were approximately twofold higher in the clearing. Higher investment of photosynthetic apparatus per unit area results in a higher LMA and $P_{g\_max\_area}$ [24]. Additionally, a greater leaf thickness increases the internal surface area available for the diffusive transfer of $CO_2$ within a leaf [24,28,82–85]. Our results therefore confirm the findings of previous studies on other species that within-species variation in LMA and leaf thickness explain the variation of area-based photosynthetic traits across different light environments [4,28,29,34,35,45,86]. The lower LMA and lower photosynthetic capacity of shade-acclimated leaves incur a lower carbon cost [23–25,27,29,82] and lower nitrogen costs [31,37,41,42,87–89] to produce a unit area of leaf. In the case of this species, having lower LMA associated with lower $P_{g\_max\_area}$ in the understory may therefore have reduced leaf construction cost per unit area, in support of the cost-saving hypotheses [24,25,81]. Such reduced LMA, or equivalently, increased leaf area per unit mass (specific leaf area, SLA), increases whole-plant leaf area and light capture with a given amount of resources as a method of acclimation to low-light environments [5,6,18,19].

### 4.2. Morphological Acclimation

Leaf laminae in the shaded understory were flatter and therefore more horizontally displayed, whereas laminae in the clearing were more upright to decrease excessive irradiance and maximize leaf area per unit ground (Figure 5b). Similar changes in leaf three-dimensional structures (i.e., flatter in the shade) to maximize light capture in low light environments have been reported for a different species of a forest herb [55], for other within-canopy variation of lamina morphology for broadleaved trees [36,67,68], and for the three-dimensional arrangement of conifer needles [90–92]. This result is consistent with several previous findings that the leaves in well-lit places are more vertically upright, while leaves in shaded places are more horizontally displayed to maximize light capture [36,45–47,60]. However, we did not focus on the consequence of the morphological acclimation in the present study. As predicted by the game theory [93], leaf angle is determined not only on the basis of optimal light capture but also on the competition [93] and/or the contact [94] with neighboring plants. Further study is therefore needed to evaluate the consequence of morphological acclimation by taking the existing competition into consideration. Additionally, in the present study, we made a simplified assumption that the leaf 3D structure was approximated by a cone (Figure 1c). However, the shape of the actual leaves was much more complex and was trumpet-shaped; a lamina was more horizontal at the edge of each leaf and gradually was more vertical towards the center (see photographs in Figure 1). In the present study, we measured photosynthetic rate only at the edge of each leaf with the LI-6400 (Figure 3); environmental heterogeneity within a single leaf [95] was not investigated. More detailed studies that model complex 3D structure [68,95] are needed for this species.

Our study had several additional limitations. First, we examined only leaves. Although investigating leaves is equivalent to investigating the entire above-ground part of individual ramets for this species (Figure 1), the importance of the whole-plant carbon economy, including roots, has long been recognized [25]. Although leaf respiration rate is positively correlated with the respiration rates of roots [96] and the entire plant [97], further studies on whole-plant respiration rates [97,98] and whole-plant biomass allocation patterns [5,6,18,20,99] are needed for this species. Second, we ignored photosynthetic induction time and instead estimated the instantaneous photosynthetic rate using steady-state photosynthetic light response curves. Efficiency of photosynthesis may differ between steady-state and short-sunfleck conditions [49,56,58] due to stomatal [42,100–106], mesophyll [42], and biochemical [102,104,107] limitations. In our dataset, however, understory leaves frequently received sunflecks during the day (Figure 9b). Leaves of forest understory plants that are induced once maintain an induced condition for a relatively long time [49,58], so the magnitude of this overestimation might not be very large. Induction times were reported to be similar between shade-tolerant and shade-intolerant species [108]. Currently, however, there is little information on within-species differences in induction time between sunlit and shaded leaves. Third, the diurnal course of the photosynthetic rate depends not only on light but also on other environmental factors (i.e., humidity,

temperature, VPD, etc.) [42,109,110] in addition to whole-plant water availability [4,111–115]. Therefore, the effect of midday depression due to stomatal closure [60,110,112,115] and photoinhibition [60,112] also would significantly alter the daily carbon gain of leaves. Furthermore, the strength of such effects may differ between sun and shade leaves [60,112,115]. Further detailed studies are therefore needed to reconfirm our findings before generation.

## 5. Conclusions

*Petasites japonicus* subsp. *giganteus* had a high capacity for acclimation to different light environments. In this species, having lower LMA associated with a lower photosynthetic rate in the understory did not increase net daily photosynthesis on the sunny day due to frequent sunflecks, but instead reduced construction costs per unit leaf area. These results indicate that when sunflecks were present, information on static leaf photosynthetic traits may not be sufficient to evaluate shade acclimation in forest understories.

**Supplementary Materials:** The following are available online at http://www.mdpi.com/1999-4907/11/12/1365/s1. Supplementary Materials: data and PPFD_and_P_net.

**Author Contributions:** Conceptualization, K.K.; methodology, K.K.; formal analysis, K.K.; investigation, R.D. and K.K.; writing—original draft preparation, R.D. and K.K; writing—review and editing, K.K.; supervision, K.K. All authors have read and agreed to the published version of the manuscript.

**Funding:** This work was funded by the Japan Society for the Promotion of Science (KAKENHI Grant Number 18K06406).

**Acknowledgments:** We thank Yasuyo Nagase and Miro Harada for performing preliminary studies. We thank staff members of Obihiro City Office and Obihiro Forest Hagukumu for permitting us to perform the fieldwork at the study sites.

**Conflicts of Interest:** The authors declare no conflict of interest.

**Data Availability Statement:** The datasets used in this article, including all the LI-6400 data, are available in the Supplementary Materials.

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
