# Peer review of "Photosynthetic and Morphological Acclimation to High and Low Light Environments in Petasites japonicus subsp. giganteus"

_forests, doi:10.3390/f11121365_

Round 1
Reviewer 1 Report
Authors investigated the photosynthetic performance (C assimilation) of a perennial herb in the filed (clearing vs. understory site). Leave morphologies were and the incoming light were assessed. No other environmental variables were included (e.g. water or nutrient regime). Authors suggest that further studies on leaf photosynthesis should also consider dynamic changes of the light regime (e.g. sunflecks).
Overall, the study reads well and the figures are nice. I like that the authors are aware of the limitations of their study and discussed them critically.
The text could be shortened by omitting unnecessary info (see below). I suggest re-arranging the discussion section to make it more exciting.
Few comments as follows ("L" refers to lines in the manuscript):
L. 38 Delete one of the “open”
L. 51 Please explain what is meant by “within-plant light gradient”? Different light climates in different tissues (e.g. epidermis is exposed to more light than tissues below)?
L. 54 Confusing formulation. Both shade leaves AND sun leaves in low light environments?
57 How can sunflecks be assigned to a diurnal fluctuation pattern? Don’t they occur highly dynamically? Sun goes up: light; sun goes down: dark. That’s diurnal. But sunflecks may change all the time during the light period.
61 Delete “e.g.,”
L 253 Did the authors evaluate differences in leaf anatomy (e.g., cross-sectioning; clearing vs. understory)? Is a certain tissue thicker in clearing leaves (e.g. chlorenchyma) and thus mainly responsible for the increased leaf thickness?
L 340 Authors show nicely that understory leaves are more horizontally orientated when compared with clearing site leaves. But I suggest not starting the discussion with this finding, as it “only” confirms textbook knowledge (leaf orientations in response to the light climate).
L 350 Same as above – this paragraph “only” confirms long-term knowledge. I do not want to denigrate the authors findings in any sense, however, I think the discussion would read more exciting by starting with the more innovative aspects of the study (e.g. discussing the role of sunflecks for the photosynthetic performance).
L. 386 What are “enzymatic costs”?
Further comments:
Authors may attempt to describe their methods clearer/more concise. Some sentences are awkwardly worded (e.g. L. 167 „The other sensor was (…)”) and some sentences can be omitted as they describe self-evident experimental design (e.g. L. 167 “Care was taken…”).
An example: “On July 4 and 7, 2020, we sampled a small, square-shaped piece (ca. 40 cm2) from the edge of each lamina with a pair of scissors for a total of 41 leaves (clearing: n = 20, understory: n = 21). The measurements of thickness and the scanning were conducted at the study site with a laptop computer immediately after sampling.” This could become something like:
“On July 4 and 7 2020, we sampled the lamina edge (ca. 40 cm2 squares) of 41 leaves (clearing: n = 20, understory: n = 21) and measured and scanned them immediately.”
Author Response
Reply to Reviewer 1
R1-1 (1st comment from Reviewer 1)
Overall, the study reads well and the figures are nice. I like that the authors are aware of the limitations of their study and discussed them critically.
R1-2
The text could be shortened by omitting unnecessary info (see below). I suggest re-arranging the discussion section to make it more exciting.
&R1-10
L 350 Same as above – this paragraph “only” confirms long-term knowledge. I do not want to denigrate the authors findings in any sense, however, I think the discussion would read more exciting by starting with the more innovative aspects of the study (e.g. discussing the role of sunflecks for the
(photosynthetic performance).
(Reply)
Per your advice, we revised the Discussion section as follows:
(1) We re-arranged this section so that it now begins with a discussion about the role of sunflecks for photosynthesis in the understory.
(2) We added the following references related to the role of sunflecks for photosynthesis in the understory.
Campany et al. 2016. Plant Cell Environ, https://doi.org/10.1111/pce.12841.
Hartikainen, S.M. et al. 2020. doi:10.3389/fpls.2019.01762.
Matthews et al. 2020. J Exp Bot, doi:10.1093/jxb/erz563.
Slattery, R.A. et al. 2018. doi:10.1104/pp.17.01234.
Wang et al. 2020 Forests https://doi.org/10.3390/f11080844.
Way & Pearcy 2012 Tree Physiol doi:10.1093/treephys/tps064.
R1-3
L. 38 Delete one of the “open”
(Reply) We deleted one of the “open”s in this sentence. Thank you for correcting the grammatical mistake.
R1-4
L. 51 Please explain what is meant by “within-plant light gradient”? Different light climates in different tissues (e.g. epidermis is exposed to more light than tissues below)?
(Reply) We apologize for the misleading sentence. We do not refer to within-leaf light gradient there. Instead, we discuss the physiological acclimation of “sun leaves” at a well-lit (usually upper) part of a canopy and “shade leaves” at a shaded (inner) part of the same canopy or of the same plant. For clarification, we revised the sentence as follows:
Analogous leaf acclimation to different light environments also has been reported for sunlit and shaded leaves within a single canopy or within a single plant.
R1-5
L. 54 Confusing formulation. Both shade leaves AND sun leaves in low light environments?
(Reply) For clarification, we revised the sentence as follows:
It has been frequently suggested that the net daily carbon gain would increase by lowering the LCP in the shade.
R1-6
57 How can sunflecks be assigned to a diurnal fluctuation pattern? Don’t they occur highly dynamically? Sun goes up: light; sun goes down: dark. That’s diurnal. But sunflecks may change all the time during the light period.
(Reply) Thank you for finding this misleading sentence. We have revised it as follows:
In the forest understory, light intensity changes diurnally due to the diurnal elevation of the sun and fluctuates dynamically due to sunflecks.
R1-7
61 Delete “e.g.,”
(Reply) We have deleted “e.g.” Thank you for this suggestion.
R1-8
L 253 Did the authors evaluate differences in leaf anatomy (e.g., cross-sectioning; clearing vs. understory)? Is a certain tissue thicker in clearing leaves (e.g. chlorenchyma) and thus mainly responsible for the increased leaf thickness?
(Reply) We completely agree that those values are very useful for readers. Unfortunately, we did not measure them. In this study, we present the physiological (photosynthetic) and morphological (3D leaf structure) data as the main findings in the Results, and we discuss anatomical acclimation only in the Discussion section.
R1-9
L 340 Authors show nicely that understory leaves are more horizontally orientated when compared with clearing site leaves. But I suggest not starting the discussion with this finding, as it “only” confirms textbook knowledge (leaf orientations in response to the light climate).
(Reply) Accordingly, we have moved those sentences from the beginning to the second subsection of the Discussion section. Additionally, we added the following references about the significance of leaf angle to support our argument and to clarify the limitations of the present study:
Anten et al. 2016, http://dx.doi.org/10.1007/978-94-017-7291-4_13
de Wit et al. 2012, doi/10.1073/pnas.1205437109.
Niinemets et al. 2006, https://doi.org/10.1086/497845
Posada, J.M et al. 2009, doi:10.1093/aob/mcn265.
Saudreau, M et al. 2017 https://doi.org/10.1111/pce.13026.
Valladares & Pearcy 2002 doi.org/10.1046/j.1365-3040.2002.00856.x
R1-11
L. 386 What are “enzymatic costs”?
(Reply) We changed “enzymatic costs” to “nitrogen costs.” The meaning of this sentence is that photosynthetic capacity is positively correlated with leaf nitrogen content per unit area (e.g., Campany 2016; doi:https://doi.org/10.1111/pce.12841; Vincent 2002 J Trop Ecol 17: 495-509. See References.), such that high photosynthetic capacity requires a higher nitrogen investment per unit leaf area.
R1-12
Authors may attempt to describe their methods clearer/more concise. Some sentences are awkwardly worded (e.g. L. 167 „The other sensor was (…)”) and some sentences can be omitted as they describe self-evident experimental design (e.g. L. 167 “Care was taken…”).
(Reply) We deleted “Care was taken…” and also revised the Methods section for brevity (see below).
R1-13
An example: “On July 4 and 7, 2020, we sampled a small, square-shaped piece (ca. 40 cm2) from the edge of each lamina with a pair of scissors for a total of 41 leaves (clearing: n = 20, understory: n = 21). The measurements of thickness and the scanning were conducted at the study site with a laptop computer immediately after sampling.” This could become something like:
“On July 4 and 7 2020, we sampled the lamina edge (ca. 40 cm2 squares) of 41 leaves (clearing: n = 20, understory: n = 21) and measured and scanned them immediately.”
(Reply) Per your request, we have revised this sentence and the entire Methods section for brevity. Thank you very much for your detailed advice and constructive comments.

Reviewer 2 Report
Sun/shade adaptation of plants has been a central topic in plant ecophyiology for a long time and the anatomical, morphological and physiological traits of shade and sun plants have been thoroughly reviewed. Therefore the present manuscript will not widen the knowledge in this field very much, except for the species-specific relations.
The authors focus on estimation of photosynthetic acclimation capacity of Petasites japonicus subsp. giganteus in different light environments, forest understory and clearing. All field measurements were performed in summer of 2020 although the seasonal course of photosynthetic activity would also be interesting especially in the understory of deciduous forest.
I miss description of microclimate conditions in the two sites, since beside light intensity, other factors also have significant influence on photosynthesis activity.
Since LI-6400 equipment was used, I suggest to include other gas exchange parameters too (e.g. stomatal conductance, transpiration rate). Leaf water relations are expected to differ in clearing from the understory that also influence the Pn. Have you made any measurements in this respect? These data, together with the results of daily microclimate measurements are especially important when daily leaf carbon gain are calculated and compared in these different sites.
Author Response
Reply to Reviewer 2
R2-1 (1st comment from Reviewer 2)
I miss description of microclimate conditions in the two sites, since beside light intensity, other factors also have significant influence on photosynthesis activity.
& R2-2
Since LI-6400 equipment was used, I suggest to include other gas exchange parameters too (e.g. stomatal conductance, transpiration rate). Leaf water relations are expected to differ in clearing from the understory that also influence the Pn. Have you made any measurements in this respect? These data, together with the results of daily microclimate measurements are especially important when daily leaf carbon gain are calculated and compared in these different sites.
(Reply) We fully agree that climatic variables other than light significantly influence photosynthetic activity. However, because we only measured diurnal course of light intensity in the present study, we do not have the data of the diurnal courses of other climatic variables at the study site.
Both stomatal conductance (Gs) and transpiration rate (Tr) were recorded simultaneously with photosynthetic rate by the LI-6400. Given your advice, we analyzed the conductance and transpiration rate and found that both the transpiration rate and stomatal conductance (measured at PPFD = 2000) were significantly (approximately twofold) higher for the sunlit leaves than the shaded leaves (p < 0.01). Nonetheless, we would rather not like to make new figures to demonstrate those findings because they are stated in the main text. The reason for this is as follows:
We conducted a literature survey on how stomatal conductance and transpiration rate were measured and analyzed in published studies. Previous studies have shown that both Gs and Tr are highly dependent on humidity or vapor pressure deficit (VPD) at the time of measurement. For this reason, it seems to us that researchers investigate transpiration rates by one of the following methods:
(1) Measure transpiration/stomatal conductance under strictly controlled humidity and temperature:
-dos Santos et al. 2020 For Ecol Manage, doi: 10.1093/jxb/erx314.
-Ouyang et al. 2017 J Exp Bot, doi: 10.1093/jxb/erx314.
-Kimura et al. 2020 J Exp Bot, doi:10.1093/jxb/eraa090.
(2) Measure the dependence of Gs/Tr on VPD:
-Campany et al. 2016 Plant Cell Environ. doi: 10.1111/pce.12841.
-Duursma et al. 2013 Agr For Meterol. doi: 10.1016/j.agrformet.2012.09.005.
(3) Measure actual diurnal fluctuation of Gs/Tr in the field:
-Espadafor et al. 2017 Sci Hort. doi: 10.1016/j.scienta.2017.06.028.
In the present study, we did not strictly control humidity and VPD and only kept favorable conditions for photosynthesis (high humidity, low VPD, morning hours of cloudy days, and wet soil after rainfall), as described in the manuscript. Additionally, we did not measure the diurnal course of Gs or Tr; we only measured the diurnal course of PPFD on the leaves. Of note, the photosynthetic rate is less dependent on humidity or VPD within a range of favorable conditions (Duursma et al. 2013, shown above). We believe that this procedure is used as a standard to compare the photosynthetic light response of leaves. However, given the studies (described above) that focused on stomatal conductance and respiration and that we did not strictly control humidity during our measurements, we cannot confidently discuss that “we found differences in stomatal conductance and transpiration between sun and shade leaves”.
Nonetheless, because we agree with you that these data are important and useful for readers who wish to use these values for different purposes, we now show all the data recorded by the LI-6400 device (including Gs, Tr, VPD, humidity, temperature, etc.) in the Supplementary Materials, so that readers can refer to those values when needed.
For these reasons, we decided not to demonstrate the observed difference of Gs and Tr between sunlit and shaded leaves in the main text for this manuscript revision. If, however, Reviewer-2 maintains (preferably with an additional rationale) that new figures containing these values are necessary in the main text, we would be happy to consider including these two values in the main text in a future revision.

Reviewer 3 Report
Dear Author
General comments
I have read the manuscript entitled: Photosynthetic and morphological acclimation to high and low light environments in Petasites japonicus subsp. giganteus (Manuscript ID: Forest -1002929) by Ray Deguchi et al. In this study, author demonstration of the acclimation capacity of the of perennial herb, Petasites japonicas in different light intensities and quantify the effect of share for the carbon gain. Author did test of this species in sun and shaded leaves insitu on both a sunny and an overcast day to see the morphological and photo synthetically acclimation. The concept of the work and whole of the research approach seems like novel and good idea and Overall the manuscript also written but author should be more careful for concise the text and highlight your findings and flow of the story. Author should consider some important points in the introduction and discussion and some related content which I mention in below. Now I request this articles in MAJOR REVISION.
Major comment
1) Line 36.
Author should improve the introduction the sense of highly matching literature citing (see below) and mention the appropriate related text. When I cross check some of the literature seems like not matching in light level of different canopy position its impact in physiology and morphology, however rest of the writing is okey in the introduction section. Please describe the light level according to plant height (lower, middle and upper) and as the width (exterior, middle and interior) and impact based on the previous studies. Some suggestions I mentioned in below.
2) Line 69-70
Without appropriate literature and questions or hypotheses entirely text make the unclear. You did the research related to the light. Moreover, you also checked light effect on the morphology (eg. LMA). You mentioned two objectives in the last section (line 69-70), please add one more objective related to light effect on morphology and accordingly please add the required information to address this third objective throughout the manuscript. That make your study increase the strengthen.
The specific comments and minor suggestions to the author are following.
1) Line 9
Author should write abstract some concrete way, please follow the pattern of some previously issued articles. Please remove the subtitle in the abstract (eg. highlight, background, result and summary and conclusions) what you wrote in separately.
Please also remember that abstract should be more solids and it should have reflected the whole experiment and writing should be concisely. Please consider these points in abstracts.
2) Line no. 13-15
“however, may not be sufficient to evaluate the effects of such acclimation on leaf carbon gain, because the light environment changes diurnally due to sun flecks in forest understories” …please this text not write in abstract it is your background of the study.
3) Line no. 22
“between the sites” this phrase makes more confuses, that means are you did in different site, I think you did the sun and shaded leaves insitu on both a sunny and an overcast day. This sentences may reflect the different site? Please revised the text.
4) Line no. 24-25
The like no 23-24 are very good but suddenly you wrote “which is consistent with previous studies on different species”, it not the good to compared to others studies in the abstract please modify the text and this related content please mentioned in discussion.
5) Line no 33
Key words seems to be more please remove very few which are less priority in your experiment.
Introduction
6) Line no 47-50
Author still little lack of the detail description of relevant information because you are testing the light effect of different canopy on the morphology and physiological perspective of herb Petasites japonicas and its responses into the shade. Therefore, please mentions the literature related to the canopy position. For your introduction this article more suitable for citation and mention the information. Article: DOI 10.1007/s13580-017-0375-y (entitled: summer pruning and reflective film enhance the….) and please describe the canopy position and available light information (interior less light, and exterior more light) then that light availability connects with the physiological traits (e.g. Photosynthesis).
(7) Material and methods are impressive
(8) Discussion:
Section 4.2 (Line 350)
Generally, to increase the physiological performance there should to be increase the leaf thickness and and increase the Chl content and increased the LMA because both help to capture the better light and higher amount of light and perform the photosynthesis better by changing the light energy into chemical energy this article very well describe the leaf thickness and chlorophyll content in response to physiological traits (photosynthetic response) very well, refer this articles https://doi.org/10.1016/j.scienta.2017.12.006, Volume 231(73-81). Entitle: Comparisons of physiological and anatomical characteristics between two cultivars in bi-leader apple trees (Malus×domestica Borkh.). Include this citation in 4.2 by mentioning the above information.
(9) Section 4.3
Carbon gain and saving (Line 360)
The flow of this section is good and informative but if you mention the information related to light effect on leaf structures and its physiological properties of evergreen tree and deciduous tree will be quite informative and strong in this section. Please consider and cite this potencial literature and please remove the of those which have less matched. Artice: Oecologia (2012) 170:11–24 DOI 10.1007/s00442-012-2279-y Entitle: Responses of leaf structure and photosynthetic properties to intra-canopy light gradients: a common garden test with four broadleaf deciduous angiosperm and seven evergreen conifer tree species.
10) Line 359: References
Double-check the citation, is all citation inside the manuscript may mention in reference? mange it formats and pattern and style of referencing if lacking, correct the language error throughout the manuscript. I hope I will have received your revised version that will significantly improve one.
Author Response
R3-1 (1st comment from Reviewer 3)
(Introduction)
1) Line 36. Author should improve the introduction the sense of highly matching literature citing (see below) and mention the appropriate related text. When I cross check some of the literature seems like not matching in light level of different canopy position its impact in physiology and morphology, however rest of the writing is okey in the introduction section. Please describe the light level according to plant height (lower, middle and upper) and as the width (exterior, middle and interior) and impact based on the previous studies. Some suggestions I mentioned in below.
& R3-6
3) Line no. 22
“between the sites” this phrase makes more confuses, that means are you did in different site, I think you did the sun and shaded leaves insitu on both a sunny and an overcast day. This sentences may reflect the different site? Please revised the text.
(Reply) We apologize that our original manuscript was confusing for readers. This study compared different individuals from two different sites (a well-lit clearing vs. a shaded understory) on a sunny day and an overcast day. Therefore, the main focus of the present study was not a comparison between the sunlit leaves from the upper canopy vs. the shaded leaves from the inner canopy. Note that the words “sun leaf” vs. “shade leaf” also are used to refer to a comparison between sites (e.g., Ref. 60: Valladares & Pearcy 2002. doi: 10.1046/j.1365-3040.2002.00856.x).
To avoid this confusion, we revised the manuscript as follows:
(1) We clarified the text of the Introduction to distinguish the previous studies related to “plants grown in well-lit vs. shaded places” from those that are related to “leaves in the sun or shade within a single canopy”, to avoid confusion.
(2) We revised the Abstract to address all the points suggested by Reviewer 3 (see below).
R3-2
2) Line 69-70
Without appropriate literature and questions or hypotheses entirely text make the unclear. You did the research related to the light. Moreover, you also checked light effect on the morphology (eg. LMA). You mentioned two objectives in the last section (line 69-70), please add one more objective related to light effect on morphology and accordingly please add the required information to address this third objective throughout the manuscript. That make your study increase the strengthen.
(Reply) Thank you very much for this advice. As suggested, we added “morphological acclimation” to the objective. To make the Introduction concise, we combined the morphological and photosynthetic acclimation into one sentence. Three objectives were therefore combined into two.
(Abstract)
R3-3
1) Line 9
Author should write abstract some concrete way, please follow the pattern of some previously issued articles. Please remove the subtitle in the abstract (eg. highlight, background, result and summary and conclusions) what you wrote in separately.
& R3-4
Please also remember that abstract should be more solids and it should have reflected the whole experiment and writing should be concisely. Please consider these points in abstracts.
R3-5
2) Line no. 13-15
“however, may not be sufficient to evaluate the effects of such acclimation on leaf carbon gain, because the light environment changes diurnally due to sun flecks in forest understories” …please this text not write in abstract it is your background of the study.
(Reply) Per your advice, we removed all subtitles from the Abstract. We had included them because they are strongly encouraged in the Author Guidelines of the journal (https://www.mdpi.com/journal/forests/instructions). However, we now explain in the Cover Letter to the Editor that we would like to remove these subheadings, as we agree with you that they are not needed. Nonetheless, if the Editor requests for us to use these subheadings, we would include them in a subsequent revision. We are grateful for your suggestion, which has helped to make the Abstract more concise.
Additionally, per your advice, we revised the Abstract for conciseness. If possible, we would like to retain “may not be sufficient to evaluate the effects of such acclimation on leaf carbon gain”, as it is the main background and objective of the present study. Although we removed the subheading (“Background and Objective”), some background we think is necessary in the Abstract. We are grateful for your detailed comments.
R3-7
4) Line no. 24-25
The like no 23-24 are very good but suddenly you wrote “which is consistent with previous studies on different species”, it not the good to compared to others studies in the abstract please modify the text and this related content please mentioned in discussion.
(Reply) (1) Per your advice, we reserved such text for the Discussion.
(2) We would like to retain the sentence, as we believe it is important to clarify which part of our results are novel. This part is a confirmation of the previously reported findings (although our finding is the first for the present study species). We conducted a literature survey and found some publications mentioning confirmation of previous studies in the Abstract, for example:
“reported previously also exists”
Wyka et al. (2012) Oecologia doi: 10.1007/s00442-012-2279-y
“Our results agree with those from other studies”
Bornmann (2010) PloS One. doi: 10.1371/journal.pone.0013345
R3-8
5) Line no 33
Key words seems to be more please remove very few which are less priority in your experiment.
(Reply) We appreciate this advice. We removed “temperate forest” from the keyword list. Note that, as described in our reply to other comments, this study compared different individuals from two different sites (well-lit clearing vs. shaded understory, or two different “light regimes”) in a temperate forest. Therefore, it is about the photosynthetic and morphological acclimation (“phenotypic plasticity”) or shade acclimation as a mechanism of “shade tolerance.” The morphological acclimations we reported include “leaf thickness,” “leaf angle,” and “leaf three-dimensional structure.” We therefore believe that the remaining keywords are relevant to the present study.
R3-9
Introduction 6) Line no 47-50
Author still little lack of the detail description of relevant information because you are testing the light effect of different canopy on the morphology and physiological perspective of herb Petasites japonicas and its responses into the shade. Therefore, please mentions the literature related to the canopy position. For your introduction this article more suitable for citation and mention the information. Article: DOI 10.1007/s13580-017-0375-y (entitled: summer pruning and reflective film enhance the….) and please describe the canopy position and available light information (interior less light, and exterior more light) then that light availability connects with the physiological traits (e.g. Photosynthesis).
(Reply) (1) Thank you very much for informing us of this literature (Bhusal et al. 2017 Hortic Environ Biotechnol) that is relevant to our study. We now cite this paper in our revised manuscript.
We apologize that our previous manuscript was confusing for readers. As mentioned above, this study compared different individuals from two different sites (well-lit clearing vs. shaded understory). Therefore, the main focus of the present study did not compare sunlit leaves from the upper canopy and shaded leaves from the inner canopy.
Nonetheless, we agree with your advice that including literature related to the canopy position will further improve our manuscript. We now cite the following references on sunlit and shaded leaves:
-Posada et al. 2009. doi:10.1093/aob/mcn265.
-Campany. 2016. doi:https://doi.org/10.1111/pce.12841.
We also clarified these points by adding the following sentence in the Abstract:
Analogous leaf acclimation to different light environments also has been reported for sunlit and shaded leaves within a single canopy or within a single plant.
We cited the suggested article (i.e., Bhusal et al. 2017) in that sentence. Thank you very much.
R3-10
(8) Discussion: Section 4.2 (Line 350)
Generally, to increase the physiological performance there should to be increase the leaf thickness and and increase the Chl content and increased the LMA because both help to capture the better light and higher amount of light and perform the photosynthesis better by changing the light energy into chemical energy this article very well describe the leaf thickness and chlorophyll content in response to physiological traits (photosynthetic response) very well, refer this articles https://doi.org/10.1016/j.scienta.2017.12.006, Volume 231(73-81). Entitle: Comparisons of physiological and anatomical characteristics between two cultivars in bi-leader apple trees (Malus×domestica Borkh.). Include this citation in 4.2 by mentioning the above information.
(Reply) Thank you very much for informing us of a recent study relevant to ours. We now cite the suggested paper (see below):
-Bhusal, N. et al. Comparisons of physiological and anatomical characteristics between two cultivars in bi-leader apple trees (Malus × domestica Borkh.). Sci. Hortic. 2018, 231, 73-81, doi:10.1016/j.scienta.2017.12.006.
Note that paragraph 4.2 in the previous manuscript was moved to become 4.1, per the suggestion of another reviewer; the above literature is cited in 4.1.
Additionally, we found another recent article from the same group that is useful to clarify a limitation of the present study and now cite it also in the Discussion:
-Bhusal et al. (2020) Responses to drought stress in Prunus sargentii and Larix kaempferi seedlings using morphological and physiological parameters. For. Ecol. Manage. 2020; 465: 118099, doi: 10.1016/j.foreco.2020.118099.
R3-11
(9) Section 4.3 Carbon gain and saving (Line 360)
The flow of this section is good and informative but if you mention the information related to light effect on leaf structures and its physiological properties of evergreen tree and deciduous tree will be quite informative and strong in this section. Please consider and cite this potencial literature and please remove the of those which have less matched. Artice: Oecologia (2012) 170:11–24 DOI 10.1007/s00442-012-2279-y Entitle: Responses of leaf structure and photosynthetic properties to intra-canopy light gradients: a common garden test with four broadleaf deciduous angiosperm and seven evergreen conifer tree species.
(Reply) Thank you very much for informing us of this paper that is related to the present study (i.e., Wyka, et al). We now cite the suggested reference.
R3-12
10) Line 359: References
Double-check the citation, is all citation inside the manuscript may mention in reference? mange it formats and pattern and style of referencing if lacking, correct the language error throughout the manuscript. I hope I will have received your revised version that will significantly improve one.
(Reply) Thank you very much for your detailed advice. As suggested, we have addressed the following points:
(1) We have checked the reference format according to the Author Guideline. We found and corrected some errors. We are grateful to you for finding the errors in the References. Additionally, we confirmed that all the literature in the References is cited in the text (we use EndNote). We also added DOIs to each reference where available, as recommended by the Manuscript Template of this journal (Note that the DOI of Valladares et al. 2000 is correct).
(2) We had our entire manuscript re-edited by native English speakers from a language-editing company before submitting the revised manuscript. Thus, some grammatical errors were corrected.
Thank you very much for your constructive comments and suggestions, which greatly improved our manuscript.

Round 2
Reviewer 3 Report
Dear Author
I have throughly read the revised manuscript as well as responses letter made by you. Over all i satisfy with your revision. Now manuscript quality is significantly improved. Now introduction and specially discussion are more solid than before. I understood 1-2 comments you addressed in different positions, due to the address the quarries of another reviewer. However, its position is correct and logical as well.
Now the abstract is more logical and concise as compared to before. One suggestion is, do not doubt about the regards to the subtitle in the abstract. It means only that "those sub-sections you should addressed only" but without mentioning the heading. Now you did well. Thank you !
Sincerely